# Potential Anti-Inflammatory and Chondroprotective Effect of *Luzula sylvatica*

**DOI:** 10.3390/ijms24010127

**Published:** 2022-12-21

**Authors:** Juliette Cholet, Caroline Decombat, Laetitia Delort, Maël Gainche, Alexandre Berry, Clémence Ogeron, Isabelle Ripoche, Marjolaine Vareille-Delarbre, Marion Vermerie, Didier Fraisse, Catherine Felgines, Adrien Rossary, Edwige Ranouille, Jean-Yves Berthon, Albert Tourrette, Julien Priam, Etienne Saunier, Yves Troin, François Senejoux, Pierre Chalard, Florence Caldefie-Chézet

**Affiliations:** 1Unité de Nutrition Humaine, Université Clermont Auvergne, CRNH Auvergne, INRAE, F-63000 Clermont-Ferrand, France; 2Institut de Chimie de Clermont-Ferrand, Université Clermont Auvergne, Clermont Auvergne INP, CNRS, F-63000 Clermont-Ferrand, France; 3Greentech, Biopôle Clermont-Limagne, 63360 Saint-Beauzire, France; 4AltoPhyto, 7 rue des Gargailles, 63370 Lempdes, France; 5Dômes Pharma, 3 rue André Citroën, 63430 Pont-du-Château, France

**Keywords:** *Luzula sylvatica*, antioxidant, PBMCs, inflammation, osteoarthritis, chondrocytes

## Abstract

(1) Interest in the *Juncaceae* family has risen as some members have shown anti-inflammatory properties and interesting compounds. In this regard, we decided to investigate the antioxidant and anti-inflammatory properties of *Luzula sylvatica*, a *Juncaceae* not yet extensively studied, in the context of osteoarthritis. (2) The *Luzula sylvatica* Ethanol extract (LS-E) was used to test the production of reactive oxygen species (ROS) by leucocytes, the IL1β and PGE2 production by peripheral blood mononuclear cells (PBMCs), the production of EP4, and the activation of NFκB in THP-1, as well as the IL1β-activated normal human knee articular chondrocytes (NHAC-Kn) gene expression, grown in monolayers or maintained in alginate beads. (3) Organic acids, caffeoylquinic acids, quercetin and luteolin, compounds frequently found in this family were identified. The LS-E exhibited inhibited ROS formation. The LS-E did not affect NFκB activation and IL1β secretion but dampened the secretion of PGE2 by PBMCs and the presence of EP4 in THP-1. It also modulated the expression of NHAC-Kn in both models and inhibited the expression of several proteases and inflammatory mediators. (4) *Luzula sylvatica* might supply interesting antioxidant protection against cartilage damages and lessen joint inflammation, notably by decreasing PGE2 secretion in the synovial fluid. Moreover, it could act directly on chondrocytes by decreasing the expression of proteases and, thus, preventing the degradation of the extracellular matrix.

## 1. Introduction

The osteoarthritis (OA) is a disease characterized by the degradation of the extracellular matrix (ECM) and joint inflammation. Globally, it is estimated that about 10% of men and 18% of women over 60 years of age have symptoms of OA. This disease has a high impact on the quality of life, as 80% of the people affected are disabled at least in part, and 25% are no longer autonomous on a daily basis [1]. The erosion of the ECM and the induction of an OA phenotype in chondrocytes is due to an elevation of the secretion of proteases (matrix metalloproteinases (MMPs) and a disintegrin and metalloproteinases with thrombospondin motifs (ADAMTS)), inflammatory mediators (especially IL1β, PGE2, and nitric oxide), and reactive oxygen species (ROS) by the chondrocytes and other cells in the joints (leucocytes, synoviocytes) [2,3,4].

Currently, the treatment of OA is mainly aimed at reducing symptoms associated with the disease, by physical therapy and the use of nonsteroidal and corticosteroidal drugs such as dexamethasone or hydrocortisone. Often, these drugs are administered by intra-articular injection [5]. Other molecules or a combination of are also investigated, such as the use of glucosamine supplementation, chondroitin sulfate, or hyaluronic acid with chitosan [6,7]. To find anti-inflammatory and antioxidant substances, the study of plants has been very promising too, yet still under-exploited. Some secondary metabolites of plants, such as phenolic diterpenes from *Rosmarinus officinalis* and *Salvia officinalis* were able to dampen the expression of some catabolic genes in a model of primary human chondrocytes stimulated by IL1β [8].

In this context, interest in the *Juncaceae* family has risen as its members have been reported to contain flavonoids, such as quercetin, luteolin, and their derivatives, coumarins, terpenes glycerides, and phenanthrenes [9,10].

The two principal genera in the *Juncaceae* family are *Juncus* and *Luzula*, the Juncus genus being the most studied. Some of their species have shown anti-inflammatory properties, such as *Juncus effusus* or *Luzula luzoides* [11,12]. The study of some characteristic metabolites, especially luteolin and phenanthrenes, has demonstrated interesting anti-inflammatory action and/or inhibitory effects against MMPs [9,13].

We chose to investigate the *Luzula sylvatica*, a very common species in woodlands and pastures of temperate regions [14,15], but whose composition and properties have not been extensively studied so far. Our previous work on this plant already showed a good antioxidant potential as well as interesting anti-inflammatory properties in an in vitro co-culture of a fibroblast and macrophage model [16].

The present study aimed at determining, in vitro, the antioxidant and anti-inflammatory potential of the plant through the exploration of leucocytes functions, and its effect on chondrocytes in the context of OA.

## 2. Results

An ethanolic extract of aerial parts of *Luzula sylvatica* (*Luzula sylvatica* ethanol extract, LS-E) was prepared as described previously by Cholet et al. and the identification of major compounds of the extract was done in the aftermath [16]. The LC-MS chromatogram of the LS-E highlighted the presence of saccharose, organic acids (quinic acid and citric acid), caffeoylquinic acids (chlorogenic acid and cryptochlorogenic acid), ananasate (1,3-O-dicaffeoylglycerol), quercetin derivatives (quercetin-3-O-rutinoside and quercetin-3-O-glucoside), and a flavone, luteolin, and its heteroside luteolin-7-O-glucoside.

### 2.1. Inhibition of Reactive Species Generation

To assess the impact of the ethanolic extract on the production of reactive oxygen species by phorbol 12-myristate 13-acetate (PMA)-stimulated blood leucocytes, the cells were incubated with 50 µg/mL of the LS-E for up to 90 min (Figure 1). This concentration was determined by a preliminary study during which we found that 50 µg/mL was sufficient to observe an impact on ROS production without affecting cells’ viability (Appendix A). Interestingly, the effect of the extract was only mild at the beginning of the PMA-induced oxidative burst (>25% of inhibition at 5 min), but it increased over time. After 15 min of incubation, more than 50% of the ROS production was inhibited by the LS-E. Moreover, this inhibitory effect lasted in time, with around 66% of inhibition after 90 min.

### 2.2. Anti-Inflammatory Effects of LS-E in Leucocytes via the Inhibition of PGE2

There was a significant increase in PGE2 secretion in the supernatants of the PBMCs in the presence of lipopolysaccharide (LPS) compared with the control cells (Figure 2a). However, the presence of the LS-E (50 µg/mL) strongly decreased the level of PGE2 and restored it to a value close to control.

The production of EP4 receptors was not increased by the presence of LPS after one hour (Figure 3). Regardless of the presence of LPS, the LS-E dampened the quantity of the EP4 receptors detected by Western blot in the THP-1 cells.

On the other hand, the NFκB pathway did not appear to be impacted by the extract. As expected, the incubation with LPS caused the translocation of NFκB p65 into the nucleus: The presence of p65 significantly decreased in the cytoplasm of the stimulated THP-1 cells and rose in parallel in the nucleus [17]. The LS-E did not affect this effect significantly. In line with that, the extract failed to inhibit the increase in IL1β induced by LPS in the supernatants of PBMCs (Figure 2b).

### 2.3. LS-E Modulates Gene Expression in IL1β-Activated Normal Human Articular Chondrocytes

The presence of IL1β strongly induced the expression of *IL1β* and *matrix metalloproteinase 1* (MMP1) expression in normal human articular chondrocytes from the knee (NHAC-kn) grown in monolayers, by 73- and 182-fold, respectively (Figure 4). The presence of LS-E (50 µg/mL) significantly decreased the expression of *IL1β* (×0.3) and that of the proteases *MMP1* and *ADAMTS4* (a disintegrin and metalloproteinase with thrombospondin motifs 4) (×0.5 and ×0.23, respectively), but not *MMP2* (Figure 5).

The LS-E was then tested on activated NHAC-kn encapsulated in a matrix of alginate. Overall, the extract tended to decrease the expression of the selected genes (Table 1). It significantly decreased the expression of six genes: the genes coding for the chemokines CCL2 and CCL5, the protease *MMP9* and the protease inhibitor *tissue inhibitors of metalloproteinases 2* (TIMP-2), the *cyclooxygenase 2* (COX-2), and *the vascular endothelial growth factor A* (VEGFA). Although not significant, the expressions of six other genes were downwardly affected with a strong tendency (0.05 < *p* < 0.1): *IL1β* and *MMP1*, in line with what was observed in monolayers of NHAC-kn, *ADAMTS1*, *MMP3*, *sex-determining region Y–type high mobility group box 9* (SOX9), and *VEGFC*.

## 3. Discussion

So far, few compounds of *Luzula sylvatica* have been identified. Our previous work on this extract showed the presence of polyphenols, luteolin, which is a hallmark of the Juncaceae, its heteroside luteolin-7-O-glucoside, and ananasate [16]. Other studies reported the presence of ferulic acid, *p*-coumaric acid in this plant, and phenanthrenoids [12,18,19,20].

The plant was first tested for its antioxidant capacity. The deleterious role of oxidative stress and ROS in the joint, in situations of traumatic injury or osteoarthritis, has been well documented [21,22]. We observed that the LS-E was able to dampen, efficiently and durably, the production of ROS by stimulated leucocytes. With NO being an inducer of chondrocytes apoptosis in presence of ROS, *Luzula sylvatica* might, therefore, supply interesting antioxidant protection against cartilage damage [4,21]. In addition, several of the compounds identified in our extract are likely to be responsible for the inhibition of xanthine oxidase (XO), notably quercetin, quercetin-3-O-glucoside, quercetin-3-O-rutinoside, and luteolin, as they have shown good inhibitory capacities in previous studies [23,24,25]. Phenanthrenes, as well, have shown anti-inflammatory activities [26].

The NFκB activation did not appear to be impacted by the extract. In line with that, the extract had no impact on the secretion of IL1β in the supernatants of PBMCs. Yet, other authors reported that sesquiterpenes were able to have a chondroprotective effect by decreasing NFκB expression [27]. As terpenoids were found previously in *L. sylvatica* [20], a similar effect could have been expected. Because terpenoids are minor compounds of the plant, their concentration might not have been enough to observe an effect on the NFκB pathway.

Nonetheless, the LS-E was able to drastically inhibit the secretion of PGE2, a major relay for nociception, by the PBMCs [28,29]. One hypothesis is that this could be due to an inhibition of the expression and/or transcription of COX-2, as this mode of action was described for luteolin and quercetin [13,30,31]. However, we had previously observed that the LS-E did not impact the expression of COX-2 in the monocytic THP-1 cell line, whereas the secretion of PGE2 decreased [16]. Therefore, the LS-E might affect either the transcription of COX-2 or is able to inhibit its activity. The last point would be in line with previous studies showing that flavonoids, among which is luteolin, are able to inhibit COX-2 activity [32].

In addition, regarding the decrease in PGE2 secretion, a decrease in EP4 protein, one of the receptors for PGE2, was observed in the presence of LS-E in the THP-1 monocytic cell line. As the PGE2/EP4 signaling is involved in the progression of OA, notably through the promotion of abnormal angiogenesis, the inhibition of both might be of interest to tackle joint inflammation [33,34].

Finally, the extract was tested on a normal articular chondrocytes cell line, with two different models: chondrocytes in monolayers and in alginate beads. Even though the same cell line was used, the phenotype in these two conditions was quite different. In monolayers, it was shown that chondrocytes lose their original phenotype and acquire a fibroblast-like one, with increased synthesis of type I collagen, among other things. In a 3D model, they retain their mature chondrocyte phenotype and can synthetize type II collagen [35,36]. On this topic, we were also able to observe an increase in the expression of marker genes for differentiation of chondrocytes when cultured long enough in alginate beads (Appendix C). In both models, the cells were stimulated with IL1β directly, so that we could observe the effect of LS-E on chondrocytes independently from its effects on PBMCs and THP-1 as with a co-culture model, for example.

In normal human articular chondrocytes from the knee (NHAC-kn) grown in monolayers, the LS-E dampened the expression of the major protease MMP1, as well as the expression of ADAMTS4, both being upregulated in the control cells stimulated with IL1β. It also decreased the expression of IL1β, whereas no effect was observed on IL1β secretion in PBMCs. Likewise, the LS-E also tended to decrease the expression of IL1β in chondrocytes in alginate beads. This suggests that the action of LS-E did not involve a Toll-like receptor (TLR)/NFκB pathway, but rather interfered with another IL1β signalization, presumably via mitogen-activated kinases (MAPK) pathway. This would be consistent with previous studies, which found that luteolin was able to inhibit IL1β-induced ERK activation [37,38], although it is to be confirmed for our extract by a study of MAPK pathway.

In activated NHAC-kn encapsulated in a matrix of alginate, overall, the extract tended to decrease the expression of the selected genes (Table 1). It significantly decreased the expression of six genes: the genes coding for the chemokines CCL2 and CCL5, the protease MMP9 and the protease inhibitor tissue inhibitors of metalloproteinases 2 (TIMP-2), the cyclooxygenase 2 (COX-2), and the vascular endothelial growth factor A (VEGFA). Although not significant, the expressions of six other genes were downwardly affected with a strong tendency (0.05 < *p* < 0.1): IL1β and MMP1, in line with what has been observe in monolayers of NHAC-kn, ADAMTS1, MMP3, sex-determining region Y-type high mobility group box 9 (SOX9), and VEGFC. In vivo, the upregulation of MMP1, MMP3, 9, and ADAMTS1 is associated with joint inflammation and osteoarthritis [29,30], whereas IL1β, CCL2, CCL5, COX-2, and VEGF A and VEGF C gene expression upregulation is thought to participate in the development of osteoarthritis and abnormal angiogenesis that accompanies it [22,31,32].

Thus, as it dampened the expression of proteases, IL1β, and the inflammatory and angiogenic mediators associated with arthritis [33], the LS-E showed a potential chondroprotective effect by directly acting on chondrocytes. However, the LS-E also decreased TIMP-2 and slightly SOX-9 expression. Nevertheless, it did not affect RUNX2 nor collagen associated genes. This observation might be investigated further to confirm an overall protective effect of the extract.

To conclude, the various properties observed for this extract, (ROS formation inhibition, presence of antioxidant compounds, anti-inflammatory effects in both circulating cells and chondrocytes) suggest that *Luzula sylvatica is* a potential candidate to combat joint inflammation and OA. Further, some nutraceuticals with similar results in vitro on PGE2 and MMPs have shown positive outcomes, with encouraging results, in in vivo studies [7]. Though the effects of the LS-E should be investigated further, notably the impact on chondrocytes secretion, *Luzula sylvatica* seems a promising anti-inflammatory plant.

## 4. Materials and Methods

### 4.1. Preparation of the Extract and Composition

The specimens of *Luzula sylvatica* were identified and collected by A. Tourrette at Marcenat, France, in June 2017. A voucher specimen (CLF 110940) was deposited at the University of Clermont Auvergne herbarium. Dried aerial parts were powered and then extracted three times with an aqueous solution containing 80% ethanol (LS-E) for 24 h as described before [16].

The major constituents of the LS-E were determined by LC-MS (UHPLC Ultimate 3000 RSLC chain) and using an Orbitrap Q-Exactive (Thermo Scientific, Illkirch, France) with an Uptisphere C18-3 (250 × 4.6 mm, 5 µm, Interchim, Montluçon, France) column, by comparison with analytical standards (Extrasynthèse, France). Results on the composition of the extract were presented in a previous publication [16].

All the experiments were performed in accordance with relevant institutional, national, and international guidelines/legislation. All donors provided their written informed consent for the use of blood samples for research purposes under Etablissement Français du Sang contract n°16-21-62 (in accordance with the following articles L1222-1, L1222-8, L1243-4 and R1243-61 of the French Public Health Code).

### 4.2. Production of Reactive Oxygen Species (ROS) by Blood Leucocytes

Blood was collected from healthy human donors (n = 4; Etablissement Français du Sang, Clermont-Ferrand, France). Leucocytes were obtained by hemolytic shock and prepared as previously described [13]. Leucocytes were incubated with or without the LS-E (50 µg/mL) and dihydrorhodamine 123 (DHR 123, 1 μM, Cayman Chemical Company, Ann Arbor, MI, USA), and stimulated, or not, by 1 µM phorbol 12-myristate 13-acetate (PMA) for 90 min. The fluorescence intensity of the formed rhodamine 123 was recorded every 5 min for 90 min (excitation/emission: 485/538 nm) using the Spark reader (TECAN Lyon, France).

Concurrently, cells from the same donors were placed in 96-well plates (10^6^ cells/mL), incubated with the extract (10, 25, 50 and 100 µg/mL) for 24 h and then resazurin (25 µg/mL) was added to track their viability (Appendix A). Fluorescence (excitation/emission: 544/590 nm) was recorded after 2 h using the Spark microplate reader (TECAN) (Appendix A).

### 4.3. IL1β and PGE2 Production of PBMCs

Blood buffy coats were harvested from healthy human donors (n = 3 to 5 donors, Etablissement Français du Sang) and layered on a gradient of Ficoll–Histopaque 1077. After centrifugation (400× *g*, 40 min at 20 °C), the first layer of plasma was aspirated, yielding a phase of monocytes and lymphocytes (PBMCs) just above the 1.077 g/mL layer. PBMCs were washed with RPMI and centrifuged twice (5 min, 400× *g*). Cells were then suspended in supplemented RPMI (10% fetal bovine serum (FBS), 50 μg/mL gentamicin, and 2 mM glutamine (Gln)), at 106 cells/mL and distributed in a 24-well plate (1 mL per well). Cells were incubated for 24 h at 37 °C under 5% CO_2_, with or without lipopolysaccharide (LPS) (1 µg/mL, LPS O26:B26, Sigma-Aldrich, Saint-Quentin Fallavier, France) and LS-E (0 or 50 µg/mL). The PGE2 concentration in the culture media was assessed by ELISA, using the PGE2 assay kit from R&D systems (R&D systems—Bio-Techne, Lille, France). The IL1β concentration was determined by ELISA using an IL1β Human ELISA Kit (Thermo Fisher Scientific, Waltham, MA, USA).

### 4.4. Western Blot Analysis of EP4 and NFκB

The human monocytic leukemia cell line, THP-1 (American Type Culture Collection) was expanded at 37 °C in a humidified atmosphere of 5% CO_2_ in a RPMI 1640 medium (GIBCO, ThermoFisher Scientific, Waltham, MA, USA) which was supplemented with 10% FBS, 2 mM Gln, and 50 µg/mL gentamicin. THP-1 cells were seeded in 6-well plates (5 × 10^6^ cells per well) and incubated for 1 h with or without the extract (50 µg/mL) and/or LPS (1 µg/mL) (n = 4). The proteins were extracted and the determination of EP4 and NFκB p65 proteins was made by Western blotting.

For NFκB, the proteins from the cytoplasm and the nucleus were separated with NE-PER kit (ThermoFisher Scientific) and tested separately by Western blot.

The proteins (30 µg) were separated by electrophoresis in a 10% polyacrylamide gel and transferred at 4 °C to a polyvinylidene membrane (Biorad, Marnes-la-Coquette, France). Immunoblots were blocked with 0.1% TBS–Tween-20, 5% dry milk, and then incubated with a primary antibody (EP4 or NFκB p65 F-6, Santa Cruz Biotechnology, Heidelberg, Germany). After that, the immunoblots were incubated with a horseradish peroxidase-conjugated secondary antibody (m-IgGκ BP-HRP: sc516102, Santa Cruz Biotechnology, Santa Cruz, CA, USA). The reactive strips were visualized by chemiluminescence (Pierce ECL Western Blotting Substrate, Thermo Fisher Scientific). Band densities were quantified using Fiji [14]. An internal control was used (β-actin, Cell Signaling Technology, Leiden, The Netherlands) to normalize signal intensities between gels.

### 4.5. Chondrocyte Culture and 3D Modeling

Human knee articular chondrocytes (NHAC-kn) were purchased from Lonza (Lonza, Basel, Switzerland) and expanded in the recommended medium (CGM—Chondrocyte Growth Medium BulletKitTM, Lonza) at 37 °C in a humidified atmosphere of 5% CO_2_.

First, the LS-E was tested on the monolayers of chondrocytes. Cells were coated in 6-well plates (200,000 cells/wells) with CGM and incubated with or without IL1β (10 ng/mL) and the extract (50 µg/mL) for 4 h.

To test the extract with 3D modeling, between passage 3 and 8, 2.106 cells were taken and suspended in 2 mL of an alginate solution (alginic acid 12.5 mg/mL, HEPES 20 mM, NaCl 150 mM). Drops of the preparations were carefully dripped into a polymerization solution (HEPES 10 mM, CaCl_2_ 102 mM) to form beads encapsulating the cells. These beads were then collected and washed, before being seeded in 6-well plates with the recommended culture medium (Chondrocyte Differentiation Medium BulletKit, Lonza) for 10 days. The LS-E (50 µg/mL) and IL1β (10 ng/mL) were added to the culture medium for 4 h. The beads were then harvested, washed in PBS and placed in a depolymerizing solution (EDTA 55 mM, HEPES 10 mM) until complete dissolution. The cells were then washed with PBS and used for further analyses. In parallel, the gene expression of collagen II and SOX-9 were monitored, and a coloration with alcian blue was realized to ensure complete re-differentiation of the chondrocytes (Appendix C).

### 4.6. Analyses of Gene Expression

Total RNA was collected using Trizol (Invitrogen, Thermo Fisher Scientific) and treated with DNAse I (Invitrogen, Thermo Fisher Scientific) to remove genomic DNA. The Reverse transcriptase (RT) was then carried out with the MultiScribe reverse transcriptase (High Capacity cDNA Reverse Transcription Kit, Applied Biosystems, Thermo Fisher Scientific) using the StepOne apparatus (StepOne Real-Time PCR System, Applied Biosystems).

The analysis of IL1β, MMP1, MMP2 and ADAMTS4 expression in the monolayers of NHAC-kn, incubated with or without extract and with or without IL1β, was realized by qPCR with the StepOne, using SybRGreen staining reagent (Thermo Fisher Scientific). The values were obtained from four independent experiments.

For NHAC-kn encapsulated in beads, the qPCR was performed on plates designed by Applied Biosystems (TaqMan Array 96-well Fast Plates, Custom format 48) using SDS7900HT apparatus (Applied Biosystems, Thermo Fisher Scientific) with TaqMan (Applied Biosystems). The analysis was conducted on 44 genes (ADAMTS1, ADAMTS13, ADAMTS8, CCL2, CCL5, COL11A1, COL1A2, COL2A1, COL4A2, COL4A4, COL5A1, COL6A1, COL6A2, COL9A2, CXCL8, IL1B, MMP1, MMP10, MMP13, MMP2, MMP3, MMP7, MMP8, MMP9, NFKB1, NFKBIA, PTGS2, RUNX2, SMAD1, SMAD2, SMAD3, SMAD4, SMAD5, SMAD7, SOX9, TGFBR1, TGFBR2, TIMP1, TIMP2, TIMP3, TNF, VEGFA, VEGFB, VEGFC) and four reference genes (18S rRNA; GUSB; GAPDH and PGK1). The relative quantification method (RQ = 2-ΔΔCT) was used to calculate the relative gene expression with ΔΔCT = [ΔCT (sample1) − ΔCT (sample2)] and ΔCT = [CT(target gene) − geometric mean CT(reference genes)]. The values were obtained from four independent experiments.

### 4.7. Statistical Analysis

Results were expressed as mean ± standard error. Gene expression obtained with TaqMan technology was analyzed with DataAssist software (Thermo Fisher Scientific). Otherwise, all statistical analyses were performed using R software (version 3.6.1). The normality of the variables was assessed by the Shapiro–Wilk test and their homoscedasticity by Bartlett’s test. Comparisons between two groups were made by the Student test or the Mann–Whitney test when normality was rejected. Values with *p* < 0.05 were considered significant.

## Figures and Tables

**Figure 1 ijms-24-00127-f001:**
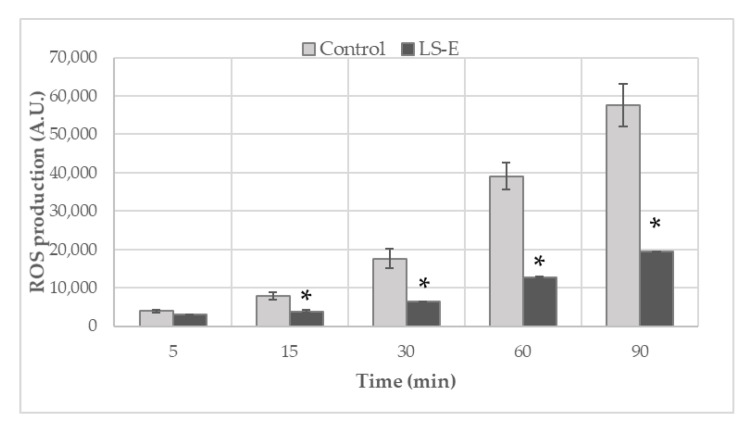
Inhibition of ROS production of blood leukocytes stimulated by PMA and incubated with or without LS-E (50 µg/mL). Results are presented as mean and standard error, n = 4, *: *p* ≤ 0.05.

**Figure 2 ijms-24-00127-f002:**
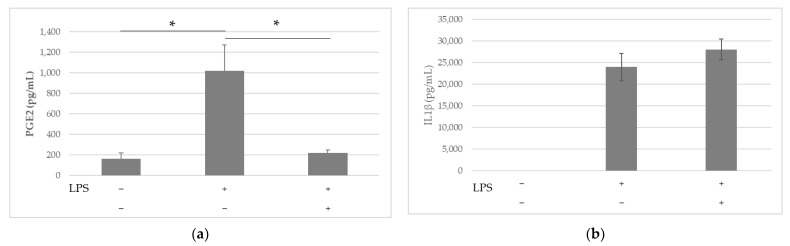
Concentration of (**a**) PGE2 and (**b**) IL1β, in the supernatants of the PBMCs incubated with or without LPS (1 µg/mL) and with or without LS-E (50 µg/mL) for 24 h; results are presented as mean and standard error, n = 5, *: *p* ≤ 0.05.

**Figure 3 ijms-24-00127-f003:**
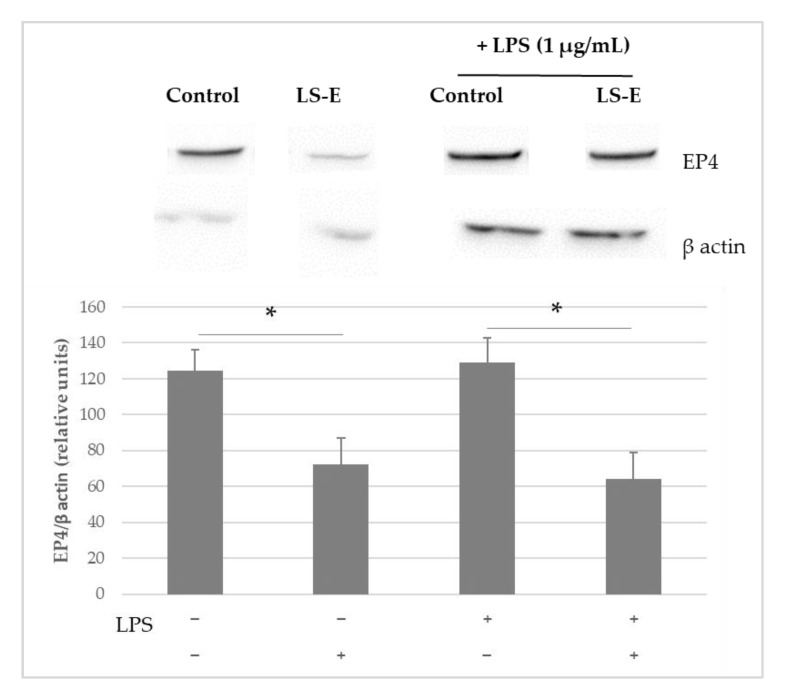
Western blot of EP4 in THP-1 incubated with LPS (1 µg/mL) and with or without LS-E (50 µg/mL) for 1 h; n = 4, *: *p* ≤ 0.05. Results were expressed as EP4/β actin ratio and control was normalized to 100%, mean values and standard error are given, n = 5.

**Figure 4 ijms-24-00127-f004:**
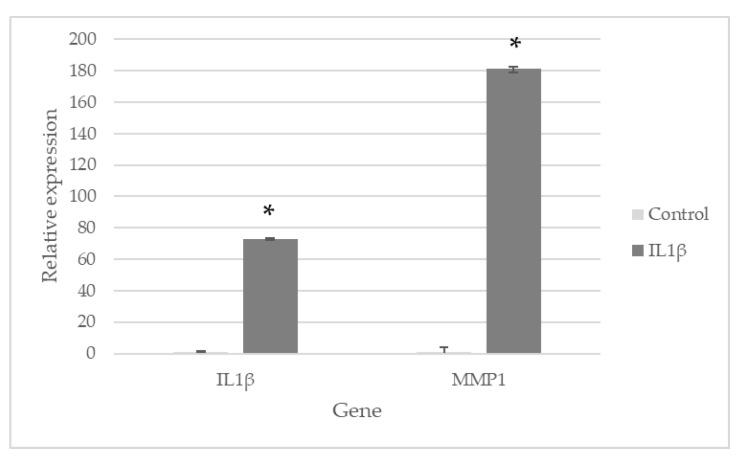
Relative gene expression (expressed as fold change) of IL1β-activated NHAC-kn in monolayers (vs. cells incubated without IL1β) n = 4. Mean values and standard error are given. *: *p* ≤ 0.05.

**Figure 5 ijms-24-00127-f005:**
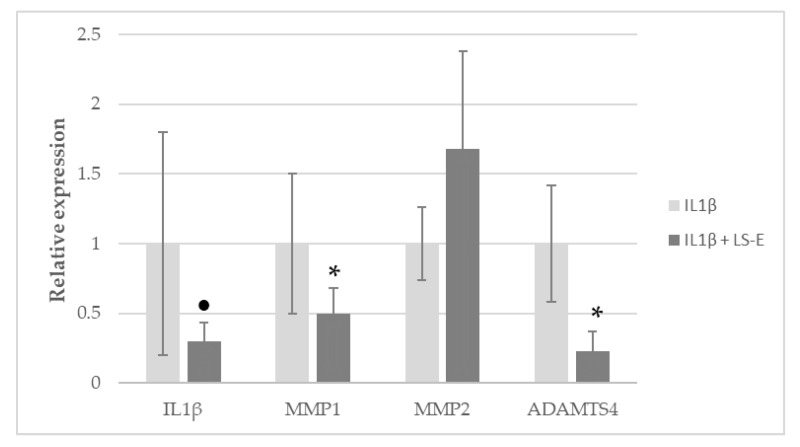
Relative gene expression (expressed as fold change) of IL1β-activated NHAC-kn in monolayers incubated with LS-E for 4 h (vs. cells incubated with IL1β and without LS-E), n = 4. Mean values and standard error are given. *: *p* ≤ 0.05, ●: 0.1 > *p* > 0.05.

**Table 1 ijms-24-00127-t001:** Relative gene expression (expressed as fold change, mean ± standard error) of activated NHAC-kn in alginate beads, incubated with LS-E and IL1β for 4 h (vs. control, cells incubated with IL1β and without LS-E). n = 4. *: *p* ≤ 0.05, ●: 0.1 > *p* > 0.05. ↓: Decrease of gene expression with LS-E (*p* < 0.1).

Gene	+LS-E 50 µg/mL Fold Change	*p* Value	
ADAMTS1	0.599 ± 0.15	0.0806 ●	↓
ADAMTS13	0.8122 ± 0.21	0.2295	
ADAMTS8	1.7912 ± 1.9	0.4708	
CCL2	0.5588 ± 0.13	0.0219 *	↓
CCL5	0.5747 ± 0.09	0.0313 *	
COL11A1	1.079 ± 0.23	0.7312	
COL1A2	1.0031 ± 0.38	0.9898	
COL2A1	0.9586 ± 0.21	0.8669	
COL4A2	0.9211 ± 0.21	0.6976	
COL4A4	0.8229 ± 0.32	0.4409	
COL5A1	0.6321 ± 0.34	0.2195	
COL6A1	0.6815 ± 0.39	0.2555	
COL6A2	0.7738 ± 0.12	0.1648	
COL9A2	0.8771 ± 0.38	0.575	
CXCL8	0.7075 ± 0.23	0.1224	
IL1B	0.4355 ± 0.42	0.0948 ●	↓
MMP1	0.5865 ± 0.17	0.0771 ●	↓
MMP10	0.6017 ± 0.23	0.2729	
MMP13	1.1403 ± 0.8	0.7569	
MMP2	0.8845 ± 0.35	0.5721	
MMP3	0.5915 ± 0.13	0.0725 ●	↓
MMP7	0.7141 ± 0.1	0.2272	
MMP8	0.8229 ± 0.32	0.4409	
MMP9	0.5816 ± 0.2	0.0326 *	↓
NFKB1	0.7712 ± 0.27	0.2674	
NFKBIA	0.9849 ± 0.18	0.9256	
COX-2	0.5002 ± 0.13	0.0339 *	↓
RUNX2	0.9286 ± 0.15	0.83	
SMAD1	0.8633 ± 0.21	0.3566	
SMAD2	0.9163 ± 0.4	0.7469	
SMAD3	0.7854 ± 0.34	0.3234	
SMAD4	0.763 ± 0.21	0.1142	
SMAD5	0.7914 ± 0.25	0.2319	
SMAD7	0.8855 ± 0.17	0.6368	
SOX9	0.648 ± 0.24	0.0676 ●	↓
TGFBR1	0.7061 ± 0.31	0.1747	
TGFBR2	0.8929 ± 0.36	0.6717	
TIMP1	1.5087 ± 1.54	0.5586	
TIMP2	0.6291 ± 0.1	0.0127 *	↓
TIMP3	0.765 ± 0.38	0.3715	
TNF	0.7866 ± 0.16	0.2445	
VEGFA	0.6396 ± 0.12	0.0274 *	↓
VEGFB	0.9229 ± 0.46	0.7932	
VEGFC	0.5561 ± 0.21	0.0514 ●	↓

## Data Availability

The data presented in this study are available on request from the corresponding author.

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
