# Peer review of "Potential Anti-Inflammatory and Chondroprotective Effect of Luzula sylvatica"

_ijms, 2022, doi:10.3390/ijms24010127_

Round 1
Reviewer 1 Report
The authors present an interesting study on the anti-oxidant, anti-inflammatory and chondroprotective properties of Luzula sylvatica in vitro.
I would recommend this manuscript for publication in the International Journal of Molecular Sciences after the following issues have been addressed:
1. In the Abstract (line 18), it is stated 'The major compounds of an ethanolic extract of Luzula sylvatica were identified'; however, the chemical composition of the extract is not mentioned at all in the Results section. Some compounds are mentioned in the Discussion (line 134-136), but there is nothing mentioned regarding their relative proportions. Section 4.1 then states that the composition of the extract was, in fact, identified in a previously published article, which was not clear from reading the abstract. Therefore, I would suggest that the sentence 'The major compounds of an ethanolic extract of Luzula sylvatica were identified' should be removed from the abstract, as this was not performed in the present work (or it should be stated clearly that this was performed in a previous study). I would also suggest a recap of the composition of the extract to be placed towards the beginning of the results section.
2. Also at the beginning of the Results section, it would be useful to the reader to say something about how the ethanolic extract was obtained/prepared. This could be something as simple as 'An ethanolic extract of LS was prepared as described previously by..... [ref].
My other issues are grammatical, typographical, or related to consistency:
3. Consistency around the equals (=) sign. For example, line 72 vs line 93 vs line 112 (not exhaustive).
4. Line 19: Luzula sylvatica should be italicised.
5. Lines 81 and 137: 'appeared' should changed to 'appear'.
6. Line 196: Use a regular hyphen instead of an em dash in 'anti-inflammatory'.
7. Line 97: '73' should be changed to '73-'
8. Line 98 and 100: Should be a multiplication sign instead of a letter x.
9. Line 54: Replace 'specie' with 'species'
10. Line 142: Replace 'we previously observed' with 'we had previously observed'.
11. Line 144: Replace 'secretion of PGE2 was decreased' with 'secretion of PGE2 decreased'.
Author Response
Thank you for your interesting comments.
Point 1: . In the Abstract (line 18), it is stated 'The major compounds of an ethanolic extract of Luzula sylvatica were identified'; however, the chemical composition of the extract is not mentioned at all in the Results section. Some compounds are mentioned in the Discussion (line 134-136), but there is nothing mentioned regarding their relative proportions. Section 4.1 then states that the composition of the extract was, in fact, identified in a previously published article, which was not clear from reading the abstract. Therefore, I would suggest that the sentence 'The major compounds of an ethanolic extract of Luzula sylvatica were identified' should be removed from the abstract, as this was not performed in the present work (or it should be stated clearly that this was performed in a previous study). I would also suggest a recap of the composition of the extract to be placed towards the beginning of the results section.
Response 1: changes have been made following this suggestion. We added a reference (Cholet, et al. Anti-Inflammatory and Antioxidant Activity of an Extract of Luzula Sylvatica in a Co-Culture Model of Fibroblasts and Macrophages. Curr Res Cmpl Alt Med 2022, 6, 2577–2201) .to the previous study as well as a summary of what was found as major compounds.
|
N |
Rta (min) |
M-Hexp (m/z) |
MS² fragment (m/z) |
Compound |
|
1 |
3.85 |
191.0547 |
191b / 85 / 127 / 173 / 59 |
Quinic acid |
|
2 |
3.93 |
341.1086 |
89b / 59 / 341 / 71 / 119 |
Saccharose |
|
3 |
6.78 |
191.0189 |
111b / 87 / 85 / 191 / 129 |
Citric acid |
|
4 |
12.97 |
353.0875 |
191b / 353 / 161 / 85 / 127 |
Chlorogenic acid |
|
5 |
14.31 |
353.0876 |
191b / 353 / 161 / 85 / 127 |
Cryptochlorogenic acid |
|
6 |
17.05 |
609.1459 |
300b / 609 / 271 / 255 / 179 |
Quercetin-3-O-rutinoside |
|
7 |
18.45 |
463.0882 |
300b / 463 / 271 / 255 / 151 |
Quercetin-3-O-glucoside |
|
8 |
19.00 |
447.0928 |
285b / 593 / 327 / 133 / 151 |
Luteolin-7-O-glucoside |
|
9 |
35.99 |
415.1031 |
415b / 253 / 161 / 179 / 135 |
Ananasate |
|
1 |
36.50 |
287.0563 |
151 b /153 / 136 / 135/ 288 |
5,7,3’,5’-tetrahydroxyflavanone |
|
11 |
38.35 |
285.0404 |
285b / 133 / 151 / 175 / 199 |
Luteolin |
|
a Retention time; b Fragment with the highest relative intensity |
||||
Point 2: Also at the beginning of the Results section, it would be useful to the reader to say something about how the ethanolic extract was obtained/prepared. This could be something as simple as 'An ethanolic extract of LS was prepared as described previously by..... [ref]..
Response 2: changes have been made following this suggestion.)
Point 3. Consistency around the equals (=) sign. For example, line 72 vs line 93 vs line 112 (not exhaustive).
Response 3: typos have been corrected.
Point 4. Line 19: Luzula sylvatica should be italicised. Lines 81 and 137: 'appeared' should changed to 'appear'. Line 196: Use a regular hyphen instead of an em dash in 'anti-inflammatory'.Line 97: '73' should be changed to '73-'. Line 98 and 100: Should be a multiplication sign instead of a letter x.Line 54: Replace 'specie' with 'species' Line 142: Replace 'we previously observed' with 'we had previously observed'. Line 144: Replace 'secretion of PGE2 was decreased' with 'secretion of PGE2 decreased'
Response 4: typos have been corrected.
Reviewer 2 Report
The article “Potential anti-inflammatory and chondroprotective effect of Luzula sylvatica “ is interesting and presents a huge and detailed work with many genes analyzed in PBMCs and chondrocytes grown both in monolayer or in a 3D structure.
It is well worth being published, but not in its current status.
here below are my comments
INTRO:
Please, rephrase and reorganize slightly adding a few additional information on other well-known corticosteroids and nonsteroidal “classical” drugs treatments (i) for OA and with other plant-based/nutraceutical treatments (ii)
here below a couple of examples for each category
exa (i): Douglas, R.J. Corticosteroid injection into the osteoarthritic knee: Drug selection, dose, and injection frequency. Int. J. Clin. Pract. 2012, 66, 699–704
exa (ii): In Vitro Effects of Low Doses of β-Caryophyllene, Ascorbic Acid and D-Glucosamine on Human Chondrocyte Viability and Inflammation. Pharmaceuticals 2021, 14, 286
exa (ii*): The Viability and Anti-Inflammatory Effects of Hyaluronic Acid-Chitlac-Tracimolone Acetonide-beta-Cyclodextrin Complex on Human Chondrocytes. Cartilage. 2020 Feb 28
Materials and Methods:
- Leucocytes were incubated with or without the LS-E (50μg/mL) μg/mL), could you please describe better the concentration/adjust the typo?
For LS-E do you mean “Luzula Sylvatica – Ethanol”? Please print it out
- What do you mean by PMA? Please print it out
- genomic RNA? (line 282) DNase I is a nuclease that cleaves DNA!
RESULTS and GRAPHS:
Fig2, b: sorry I don’t understand; the concentration of IL1B seems comparable with the not treated with the plant extract; as you claim in the lines above….so it is not effective as an anti-inflammatory?
you treated and checked in the supernatants the same molecule – IL1B, is it possible that you are observing basically the same stuff you added? How many washes did you do to the culture?
Line 159: “the cells were stimulated with IL1β directly” Have you proven that the molecule spreads freely through alginate gel?
is it possible to reprepare the graphs (fig4 e fig 5) with also the columns of the controls in the same graph? How they are presented in the current form are difficult to interpret.
perhaps try with the 2ˆdCt method, where dCt = Ct mean controls genes − Ct target gene.
The red line in the fig5 is not described in the caption, what does it mean?
with “controls” I mean: cells untreated nor with inflammatory stimulus, nor with plant extract; cells treated only with inflammatory stimulus.
RESULTS and DISCUSSION:
Could you please explain better the timing of your treatments?
How did you chose the LS-E (50 μg/mL) concentration?
Why did you quantify only few genes in the monolayer and a huge panel in the 3D culture?
Do the cells have the same characteristics? If not in what they are different?
Since in the monolayer they are more fibroblast-like (as you claim in the 156-157 lines) add some sentences to support your investigations in both monolayer or structure.
Please add a graph in which you compare two/three of the fundamental markers of differentiated chondrocytes in the monolayer and in the alginate structure to corroborate your results.
Could you please add an image(s) of these 3D structures- matrix of alginate?
Table 1: this is only a stylistic tip, to make your table more readable and fancy, I suggest adding a tiny arrow up or down to highlight if the gene increases or decreases
GENERAL:
Did you have some evaluation of the Effects on viability and/or toxicity on Chondrocyte of your product? A simple MTT with an increasing quantity of the product at different time points (1h, 6h, 24h and 48h) will be enough.
On chondrocytes, NF-κB plays a key role in inducing, in turn, cytokine expression and cytokine-induced secretion of MMPs.
Many anti-inflammatory and chondroprotective drugs have effects on NF-kB decreasing its expression along with MMP13 also in chondrocytes and increasing the protective expression of Collagen II and aggrecan (In Vitro Effects of Low Doses of β-Caryophyllene, Ascorbic Acid and D-Glucosamine on Human Chondrocyte Viability and Inflammation. Pharmaceuticals 2021, 14, 286); do you have any hypothesis on why your extract has no effects in any of them?
Do you by chance have the possibility to analyze the expression on aggrecan?
I suggest you to add in the discussion some comparisons with other “nutraceutical” studies, in addition to the abovementioned you can look to this review “Ragle, R.L.; Sawitzke, A.D. Nutraceuticals in the management of osteoarthritis: A critical review. Drugs Aging 2012, 29, 717–731.”
Author Response
Thank you for your interesting comments.
Point 1: . INTRO:
Please, rephrase and reorganize slightly adding a few additional information on other well-known corticosteroids and nonsteroidal “classical” drugs treatments (i) for OA and with other plant-based/nutraceutical treatments (ii)
here below a couple of examples for each category
exa (i): Douglas, R.J. Corticosteroid injection into the osteoarthritic knee: Drug selection, dose, and injection frequency. Int. J. Clin. Pract. 2012, 66, 699–704
exa (ii): In Vitro Effects of Low Doses of β-Caryophyllene, Ascorbic Acid and D-Glucosamine on Human Chondrocyte Viability and Inflammation. Pharmaceuticals 2021, 14, 286
exa (ii*): The Viability and Anti-Inflammatory Effects of Hyaluronic Acid-Chitlac-Tracimolone Acetonide-beta-Cyclodextrin Complex on Human Chondrocytes. Cartilage. 2020 Feb 28
Response 1: This is indeed worth to be reminded. A small paragraph has been added to the introduction.
Point 2: MATERIALS AND METHODS:
Leucocytes were incubated with or without the LS-E (50μg/mL) μg/mL), could you please describe better the concentration/adjust the typo? For LS-E do you mean “Luzula Sylvatica – Ethanol”? Please print it out - What do you mean by PMA? Please print it out- genomic RNA? (line 282) DNase I is a nuclease that cleaves DNA!
Response 2: The typo have been corrected.
Point 3: RESULTS and GRAPHS
Fig2, b: sorry I don’t understand; the concentration of IL1B seems comparable with the not treated with the plant extract; as you claim in the lines above….so it is not effective as an anti-inflammatory? you treated and checked in the supernatants the same molecule – IL1B, is it possible that you are observing basically the same stuff you added? How many washes did you do to the culture?
Line 159: “the cells were stimulated with IL1β directly” Have you proven that the molecule spreads freely through alginate gel? is it possible to reprepare the graphs (fig4 e fig 5) with also the columns of the controls in the same graph? How they are presented in the current form are difficult to interpret.perhaps try with the 2ˆdCt method, where dCt = Ct mean controls genes − Ct target gene.
The red line in the fig5 is not described in the caption, what does it mean? with “controls” I mean: cells untreated nor with inflammatory stimulus, nor with plant extract; cells treated only with inflammatory stimulus.
Response 3: Fig2 We treated cells with LPS (lipopolysaccharide from E. coli), thus we are quite confident about our results for IL1b measurement. Fig 4 and 5, we modified the graph so that it is easier to understand. Line 159: the diffusion of IL1b in alginate beads has been well documented (we could cite Arlov, Øystein, et al. "Biomimetic sulphated alginate hydrogels suppress IL-1B-induced inflammatory responses in human chondrocytes." (2017). Morevover when observing the difference of Ct for some genes, notably IL1b, the expression of IL1b was higher in Chondrocytes in presence of IL1b protein (Ct =30) than without it (Ct> 35; leading to a 2ˆdCt value ~72 for cells + IL1b ) . Therefore, we think we can safely conclude that IL1b was able to diffuse in alginate beads.
Point 4. RESULTS and DISCUSSION:Could you please explain better the timing of your treatments? How did you chose the LS-E (50 μg/mL) concentration?
Response 4: We added a figure in Appendix A to better explain the choice of 50µg/ml. We first worked with a methanolic extract of the plant before switching to an ethanolic extract. With this methanolic extract, we observe that 50µg/ml was sufficient to observe a significant effect on ROS production, without affecting viability.
Point 5. Why did you quantify only few genes in the monolayer and a huge panel in the 3D culture?
Do the cells have the same characteristics? If not in what they are different?. Since in the monolayer they are more fibroblast-like (as you claim in the 156-157 lines) add some sentences to support your investigations in both monolayer or structure.
Please add a graph in which you compare two/three of the fundamental markers of differentiated chondrocytes in the monolayer and in the alginate structure to corroborate your results. Could you please add an image(s) of these 3D structures- matrix of alginate?
Response 5: The results obtained for Luzula sylvatica are based on a screening of numerous plant extracts. When we first quantified genes in the monolayer, we were limited due a large amount of plants to screen. After this screening, Luzula sylvatica ranked among the most interesting and then we were able to quantify much more genes in the second part of our study with alginate beads.
In appendix C, we added some pictures we were able to make when we made alginate beads. We included also a graph with the expression of SOX9 and Collagen II genes, showing their expression increased over time in chondrocytes in beads. An Alcian Blue staining highlighted the presence of glucosaminoglycan as well.
Regarding the fibroblast like monolayer, we cited Melero-Martinet al. and Charlier et al.. and added some sentences in the discussion part.
Point 6. Table 1: this is only a stylistic tip, to make your table more readable and fancy, I suggest adding a tiny arrow up or down to highlight if the gene increases or decreases '.
Response 6: Arrows have been added
Point 7. Did you have some evaluation of the Effects on viability and/or toxicity on Chondrocyte of your product? A simple MTT with an increasing quantity of the product at different time points (1h, 6h, 24h and 48h) will be enough. '
Response 7: Unfortunately we missed that data. We have performed several viability assay on different cell lines (PBMCs, THP-1…), but this an accidental omission. However, as we investigated first the effect of the extract on monolayers of chondrocytes, we did not observe any cell detachment in the wells. We may perhaps add that the duration of incubation in these conditions was 4h, which is a relatively short to observe an effect on viability, although it does not completely excuse our omission.
Point 8 On chondrocytes, NF-κB plays a key role in inducing, in turn, cytokine expression and cytokine-induced secretion of MMPs... Many anti-inflammatory and chondroprotective drugs have effects on NF-kB decreasing its expression along with MMP13 also in chondrocytes and increasing the protective expression of Collagen II and aggrecan (In Vitro Effects of Low Doses of β-Caryophyllene, Ascorbic Acid and D-Glucosamine on Human Chondrocyte Viability and Inflammation. Pharmaceuticals 2021, 14, 286); do you have any hypothesis on why your extract has no effects in any of them? Do you by chance have the possibility to analyze the expression on aggrecan.
Response 8: We do not have unfortunately the possiblity to redo the experiment I a near future, thearefore it would be difficult to analyse the expression of aggrecan, although it would have been interseting with regards to the publication mentionned. For now we are not certain about why the extract does not act on NFkB – L. sylvatica does contain terpenoids (Gainche, et al. “Anti-Inflammatory and Cytotoxic Potential of New Phenanthrenoids from Luzula Sylvatica.” Molecules vol. 25,10 2372. 20 May. 2020, doi:10.3390/molecules25102372) and other sesquiterpene, notably α-copaene have been reported in Juncacea (Grassland Science in Europe, Vol. 19 - EGF at 50: the Future of European Grasslands). However, they might not be able to act in the same manner that β-Caryophyllene; or maybe the amount of terpenoids are not enough to observe an effect. We added some sentences in the discussion part.
Point 9. I suggest you to add in the discussion some comparisons with other “nutraceutical” studies, in addition to the abovementioned you can look to this review “Ragle, R.L.; Sawitzke, A.D. Nutraceuticals in the management of osteoarthritis: A critical review. Drugs Aging 2012, 29, 717–731.”.
Response 9: We added some sentences in the discussion part.

Round 2
Reviewer 2 Report
The authors have responded satisfactorily to all my questions/observation